# Comparison of the Mental Health Impact of COVID-19 on Vulnerable and Non-Vulnerable Groups: A Systematic Review and Meta-Analysis of Observational Studies

**DOI:** 10.3390/ijerph182010830

**Published:** 2021-10-15

**Authors:** Soo-Hyun Nam, Jeong-Hyun Nam, Chan-Young Kwon

**Affiliations:** 1Department of Nursing, College of Nursing, Seoul National University, Seoul National University Hospital, Seoul 03080, Korea; myhoi83@naver.com; 2Department of Korean Medicine, School of Korean Medicine, Pusan National University, Busan 50612, Korea; jhnam102@naver.com; 3Department of Oriental Neuropsychiatry, College of Korean Medicine, Dong-Eui University, Busan 47227, Korea

**Keywords:** mental health, COVID-19, coronavirus, pandemic, vulnerability

## Abstract

Vulnerable populations may be more vulnerable to mental health problems posed by the coronavirus disease 2019 (COVID-19) pandemic. A systematic review was performed to compare the mental health impact of COVID-19 between vulnerable and non-vulnerable groups. Five electronic databases were searched for observational studies reporting the psychological outcomes of both vulnerable populations and healthy controls during the COVID-19 era. The primary outcomes are the severity of depression and anxiety, and secondary outcomes include other aspects of mental health such as stress or sleep disturbance. Meta-analysis was performed for the severity of mental health symptoms, and the results were presented as standardized mean difference and 95% confidence intervals. A total of 25 studies were included. According to the findings, the elderly generally experienced significantly lower levels of psychological symptoms including depression, anxiety, and perceived stress. Pregnant women, patients with chronic diseases, and patients with pre-existing severe mental disorders showed mixed results according to each mental health outcome. The results indicate that vulnerable groups have been affected differently in the COVID-19 era. Though the insufficient number and heterogeneity of included studies leave the results inconclusive, our findings may contribute to identifying priorities of mental health needs among various vulnerable populations and allocating health resources with efficiency.

## 1. Introduction

The Coronavirus disease 2019 (COVID-19) pandemic has been posing a serious threat to people worldwide, not only due to its direct impact on physical health, but also due to its impact on psychological health. Overall, the primary therapeutic goal against the COVID-19 pandemic has mainly been focused on treatment for reducing mortality and prevention strategies, such as social distancing and quarantine [1], whereas mental health issues have not received much attention. However, as COVID-19 creates a huge impact on various aspects of people’s daily life, including separation from loved ones, uncertainty concerning the rapid spread of the pandemic, fear of being infected, feeling a loss of control and freedom, facing urgent socio-economic changes including job loss, financial issues, and changes in daily life style [2,3,4], one can easily assume that its negative psychological influences penetrate deeply into individuals’ daily life. Although psychological strain is difficult to notice and easily overlooked [5], considering its pervasive and accumulating effect on people compared to physical symptoms caused by the infection of severe acute respiratory syndrome coronavirus 2 (SARS-CoV-2) [6,7], the psychological impacts of COVID-19 are an important issue. According to a systematic review of general populations affected by the mental health consequences of COVID-19, the prevalence of depression, anxiety, insomnia, posttraumatic stress disorder (PTSD), and psychological distress were 15.97%, 15.15%, 23.87%, 21.94%, and 13.29%, respectively [8]. Mental health experts have also continuously expressed concern over the negative long-term psychological impact of COVID-19 [9], referring to it as ‘new wave of pandemic’ and a ‘post-pandemic wave’ [10], which alarms the detrimental aftereffect of psychological consequences during this unprecedented pandemic era [11].

With respect to mental health resources, the data from the World Health Organization (WHO)’s Mental Health Atlas 2017 demonstrated the scarcity and disparities of the resources within/between countries [12]. This suggests the necessity of efficient utilization of mental health resources to meet needs. Meanwhile, vulnerable groups have been affected differently in other pandemics. During the 2003 severe acute respiratory syndrome (SARS) outbreak, pregnant women showed higher levels of anxiety than the pre-SARS control [13]. In addition, children are likely to develop fear and anxiety-related cognitive biases from threat information given by their parents during an influenza pandemic [14]. Additionally, in other natural disasters such as earthquakes, the elderly survivors were more susceptible to PTSD and general psychiatric morbidity than their younger counterparts [15]. As the elderly are reported to belong to fewer social networks, they may lack the social capital resources available during a disaster [16]. As such, different vulnerable groups may express dissimilar weaknesses, and evaluating the specific mental health vulnerabilities of certain groups could enable the efficient utilization of mental health resources on people most in need.

Psychological burden induced by COVID-19 pandemic disproportionately affects vulnerable populations, which calls for effective and fair support for those who are at the highest risk [17]. However, little is known about how COVID-19 has differently affected the mental health of each vulnerable population. To the best of our knowledge, there is no published article that comprehensively reviews and synthesizes data on mental health among vulnerable populations and compares the mental health impact of COVID-19 between vulnerable and non-vulnerable groups. Thus, we conducted a systematic review and meta-analysis to assess psychological and mental impact among vulnerable populations during the COVID-19 pandemic compared to healthy controls, through which the priorities of mental health needs can be identified.

## 2. Materials and Methods

We followed the Preferred Reporting Items for Systematic Reviews and Meta-Analyses (PRISMA) Statement [18] (Appendix A). In addition, the protocol of this systematic review was registered in the Open Science Framework registries (osf.io/k8uy7). There were no amendments to the methodology after protocol registration.

### 2.1. Search Strategy

Two independent researchers (SH Nam and JH Nam) conducted a comprehensive search in following five electronic databases: Medline via PubMed, EMBASE via ovid, Cochrane library, PsycINFO, and Cumulative Index to Nursing & Allied Health Literature (CINAHL). The search strategy was reviewed by an experienced librarian (Appendix A). There was no restriction on publication status (including gray literature), or publication country. The study search was conducted on 5 March 2021, and all studies published from declaration of the pandemic (11 March 2020) up to the search date (5 March 2021) were considered.

### 2.2. Eligibility Criteria

#### 2.2.1. Types of Studies

Original cross-sectional or longitudinal studies reporting the psychological outcomes among vulnerable populations with healthy controls during COVID-19 were included. In the case of longitudinal studies, we included the first data measured right after the declaration of the pandemic. The following cases were excluded if they: (1) were not written in English, and (2) were abstracts, editorials, narrative reviews, opinions, perspectives, or letters.

#### 2.2.2. Types of Participants

We defined our target populations (i.e., vulnerable populations) based on previous research considering vulnerability in the COVID-19 context [19,20]. People with chronic diseases, who are particularly suggested as vulnerable individuals in COVID-19 era by the Center for Disease Control and Prevention (CDC) [21], patients with serious mental illness, including schizophrenia, bipolar disorder, obsessive-compulsive disorder, major depressive disorder, generalized anxiety disorder, and panic disorder, those with a disability (disabled), elderly people over 60 years of age, youth/young people/children under 18 years of age and pregnant women are included as vulnerable populations. In the case of youth/young people/children under 18 years of age, although the risks of severe illness and mortality caused by COVID-19 are lower for young people, these populations may be vulnerable in terms of mental health due to other factors, including the loss of education [17] as well as a lack of psychological capital to preserve their mental health [22], as a result of school closures. In addition, child abuse/maltreatment, domestic violence [17], and the unemployment of their parents [23] may affect their mental health status. People with low socioeconomic status such as the unemployed, those with low income or precarious employment, homeless, single parents, racial/ethnic minorities, immigrants, refugee groups, and underinsured (uninsured patients or patients without health insurance) were included. There was no restriction on the severity of symptoms, gender, ethnicity, or race of patients. However, studies which did not present the participants’ information of age ranges or age criteria were excluded.

#### 2.2.3. Types of Outcome

The primary outcomes are the severity of depression and anxiety diagnosed by clinicians or assessed using validated assessment tools as follows.

Depressive symptoms measured by validated tools such as Patient Health Questionnaire (PHQ-9) [24], Beck’s Depression Inventory II (BDI) [25], Center for Epidemiologic Studies Depression Scale (CES-D) [26], Depression subscale of the Depression, Anxiety, and Stress Scale-21 (DASS-21) [27], and Hospital Anxiety and Depression Scale (HADS) [28];Anxiety symptoms measured by validated instruments such as Generalized Anxiety Disorder Scale-7 (GAD-7) [29], the State-Trait Anxiety Inventory (STAI) [30], Anxiety subscale of DASS-21 [27], and HADS [28].

Secondary outcomes included perceived stress measured by Perceived Stress Scale (PSS) [31], PTSD measured by the Impact of Event Scale (IES) [32], and Primary Care PTSD Screen for DSM-5 (PC-PTSD-5) [33]; sleep quality was measured by validated assessment tools, such as the Insomnia Severity Index (ISI) [34], the Pittsburgh Sleep Quality Index (PSQI) [35], and the Leeds Sleep Evaluation Questionnaire (LSEQ) [36].

### 2.3. Study Selection

Two reviewers (SH Nam and JH Nam) independently performed a study selection to determine whether the searched studies met the inclusion/exclusion criteria. For the first inclusion, the titles and abstracts of the searched articles were assessed, and the full texts of all eligible studies were reviewed for their relevance. Disagreement regarding study selection was discussed by the two reviewers to resolve, and if the discrepancies were not resolved the corresponding author (CY Kwon) intervened.

### 2.4. Data Extraction

Data extraction was independently performed by two reviewers (SH Nam and JH Nam) using a standardized form in Microsoft Excel, 2016. The following information were extracted from included articles: first author’s name, year of publication, population (type of vulnerability), socio-demographic characteristics (country, gender proportion, age, ethnicity), study designs, total sample size, response rate, number of drop-outs, outcomes related to mental health, and the results. Disagreement regarding data extraction was discussed by the two reviewers to resolve, and if the discrepancies were not resolved the corresponding author (CY Kwon) intervened.

### 2.5. Assessment of Study Quality

The methodological quality of included studies was assessed by two independent reviewers (SH Nam and JH Nam) using the Joanna Briggs Institute (JBI) prevalence critical appraisal tool [37], which has been predominantly used in reviews on observational studies for reporting prevalence. This tool includes 9-items as follows: sample frame, sampling method, adequacy of sample size, description of the study settings and subjects, reliability and validity of measurements, appropriate statistical analysis, response rate and management of inadequate response rate. Each item was assessed and evaluated as being “yes”, “no”, “unclear”, or “not applicable” with reasons corresponding to the evaluation. Disagreement regarding the methodological quality assessment was discussed by the two reviewers to resolve, and if the discrepancies were not resolved the corresponding author (CY Kwon) intervened.

### 2.6. Data Analysis

In this review, quantitative synthesis was attempted according to mental health symptoms. Software Stata version 13.1 (StataCorp, College Station, TX, USA). and metan-code were used for the meta-analysis. A random-effect model was used, and the severity of mental health symptoms was presented as standardized mean difference (SMD) and 95% confidence intervals (CIs). I² statistic was used to estimate statistical heterogeneity, and I² values greater than 50% and 75% were interpreted as substantial and considerable heterogeneity, respectively. However, the heterogeneity of included studies, including the participants’ living environment, times from the COVID-19 outbreak, type of vulnerability, and potential comorbid diseases, were evident. Therefore, in this review, attention was paid to the values of data from individual study rather than pooled data in our meta-analyses.

### 2.7. Publication Bias

When 10 or more studies were included in each meta-analysis, publication bias was evaluated through funnel plots.

## 3. Results

### 3.1. Study Selection

Among the searched 12,382 documents, titles and abstracts of 8689 studies were screened after excluding duplicate documents. The initial screening yielded a review of the full-texts of 56 potentially eligible studies, of which 31 studies that did not meet the inclusion criteria and were excluded (Appendix A). Finally, 25 studies were included in this review [6,36,37,38,39,40,41,42,43,44,45,46,47,48,49,50,51,52,53,54,55,56,57,58,59]. All of the included studies were written in English. Among them, 17 studies [38,39,40,41,42,43,44,45,46,47,48,49,50,51,52,53,54,55] were included in the meta-analysis (Figure 1).

### 3.2. Characteristics of Included Studies

Of the 25 studies included, 17 [6,39,40,43,44,45,46,48,49,50,51,53,54,55,56,57,58] were cross-sectional studies, and of the remaining studies, 6 [38,41,42,52,59,60] were case-control studies, and 2 [47,61] were longitudinal studies. Four studies [38,44,54,59] were conducted in Turkey, three in the United States [43,46,61], three in Spain [49,50,51] and Italy [53,55,58], two in Australia [40,48], China [39,56], and Iran [41,45], and one Germany [52], the Netherlands [60], Bangladesh [42], Israel [6], and Argentina [47], respectively. 22 studies [6,38,40,41,42,43,44,45,47,48,49,50,51,52,53,54,55,56,57,58,59,60] did not report the ethnicity of the participants. Otherwise, except for one study [39] involving only Chinese participants, the rest [46,61] were multi-ethnic studies. The subtypes of vulnerable populations among included studies were chronic disease patients in eleven studies [39,41,42,44,52,53,54,57,58,59,60], elderly [6,40,46,49,51] and SMI patients [43,48,50,55,61] in five studies, and pregnant women in four studies [38,45,47,56], respectively. The sample size ranged from 48 to 13,829. Except for three studies [6,40,58] that did not report participants’ ages, three studies reported median with interquartile range [41,44,46], and one study [53] reported range of age; all the remaining studies reported mean age. The most frequently used assessment tool was DASS-21 which was used in 7 studies [42,48,49,50,51,53,55] measuring depressive symptoms, anxiety, and stress. PHQ-9 was used in 5 studies [40,46,56,57,61] for assessing depressive symptoms, GAD-7 was used in 5 studies [40,46,56,57,61] for assessing anxiety, HADS was used in 5 studies [39,41,45,59,60] for assessing both depression and anxiety, and IES was used in 4 studies [46,49,50,51] for evaluating PTSD symptoms. There were only two studies [39,61] measuring sleep disturbance which used PSQI (Table 1).

### 3.3. Methodological Qualities of Included Studies

For question 1, “*Was the sample frame appropriate to address the target population?*”, 13 studies [6,40,42,43,46,47,49,50,51,56,57,58,60] that used online surveys spread through social media or websites using organizational and personal networks, and presented three or more sociodemographic characteristics or medical history of the participants, were evaluated as Y. Eight studies [38,39,41,44,52,54,55,61] that were conducted at a single institution (a hospital department, a clinical service, etc.) were evaluated as N, while four studies [45,48,53,59] that did not mention the sample frame were evaluated as U. For question 2, “*Were study participants sampled in an appropriate way?*”, no study was rated as Y as there were no studies that performed random sampling. Eighteen studies [6,39,40,41,42,43,44,45,46,47,49,50,51,52,56,57,58,61] that did not perform random sampling were evaluated as N, and seven studies [38,48,53,54,55,59,60] in which the sampling method was not mentioned were evaluated as U. For question 3, “*Was the sample size adequate?*”, 13 studies [6,40,42,45,46,49,50,51,53,56,57,58,60] for which sample size calculations were performed or with more than 500 participants were rated as Y based on previous study [62]. Nine studies [38,39,43,44,47,52,54,55,61] that did not clarify how the sample size was calculated were evaluated as U, while three studies [41,48,59] with less than 100 participants were rated as N. For question 4, “*Were the study subjects and the setting described in detail?*”, 24 studies [6,38,39,40,41,42,43,44,45,46,47,48,49,50,51,52,53,55,56,57,58,59,60,61] describing detailed sociodemographic characteristics of participants such as age, gender, marital status, income, and education were evaluated as Y. However, Cakiroglu et al. (2020) [54], in which sociodemographic characteristics of participants were collected but not presented in detail, was evaluated as U. For question 5, “*Was the data analysis conducted with sufficient coverage of the identified sample?*”, 19 studies [6,39,40,43,44,45,46,47,48,49,50,51,53,54,55,56,57,58,61] that did not present either the response rate or drop-out rate of subgroups were evaluated as U. Six studies [38,41,42,52,59,60] that used case-control design were rated as NA. For question 6, “*Were valid methods used for the identification of the condition?*”, all included studies [6,38,39,40,41,42,43,44,45,46,47,48,49,50,51,52,53,54,55,56,57,58,59,60,61] used valid and reliable assessment tools mentioned in the Method section, and were evaluated as Y. For question 7, “*Was the condition measured in a standard, reliable way for all participants?*”, 19 studies [6,38,39,40,42,45,46,47,48,49,50,51,52,55,56,57,59,60,61] in which all participants answered the same questionnaire during the same period were evaluated as Y, while Ciprandi et al. (2020), in which two groups conducted surveys at a different time period [58], was evaluated as N. Five studies [41,43,44,53,54] that conducted verbal interviews without a detailed description of how reliability was ensured throughout the process (e.g., if the data collectors had undergone a standard training) were evaluated as U. Muro et al. (2020) was also rated as U, since it did not describe how the data were collected [53]. For question 8, “*Was there appropriate statistical analysis?*”, 24 studies [6,38,39,40,41,42,43,44,45,46,47,48,49,50,51,52,54,55,56,57,58,59,60,61] were rated as Y since they used appropriate statistical analysis, such as t-tests, one-way ANOVA, Mann–Whitney U tests, and Chi-Square tests, and presented means and standard deviations (SD). However, that of Muro et al. (2020) [53] was evaluated as N, since they did not present means and SD with appropriate statistical analysis but only presented the N(%) of the participants’ responses. For question 9, “*Was the response rate adequate, and if not, was the low response rate managed appropriately?*”, 19 studies [6,38,39,40,42,43,44,45,46,47,48,49,50,51,54,55,56,57,58] did not mention the response rate and were evaluated as U. Two studies [53,61] with more than 60% of the total response rate were evaluated as Y based on previous study [63]. Two studies [52,60] with less than 60% of the total response rate, and where non-response appeared to be related to the outcome measured as well as the characteristics of non-responders disparate to those who responded, were evaluated as N. Two studies [41,59] in which the participants were extracted through previous research were rated NA (Table 2).

### 3.4. Mental Health Impact of COVID-19 on Vulnerable Groups

#### 3.4.1. Depressive Symptoms: Primary Outcome

**Elderly:** In the context of COVID-19, elderly (65+ or 70+) groups generally showed significantly lower depressive symptoms than non-elderly groups on PHQ-9 (WMD −4.59, 95% CI: −5.45 to −3.73 (65+ vs. 18–39 age); −1.84, −2.60 to −1.06 (65+ vs. 40–64 age); −6.10, −6.64 to −5.56 (70+ vs. 18–29 age); −3.80, −4.24 to −3.36 (70+ vs. 30–49 age); −1.90, −2.33 to −1.47 (70+ vs. 50–69 age)) as well as DASS-21 Depression score (−1.77, −2.78 to −0.76 (60+ vs. 18–25 age)); (−0.97, −1.93 to −0.01 (60+ vs. 26–33 age)). In addition, there was borderline significance between elderly (60+) group and 34–45 age group (−0.61, −1.55 to 0.33) or 46–60 age group (−0.34, −1.30 to 0.62) on DASS-21 Depression score. Moreover, both male elderly (60+) (−0.30, −0.37 to −0.23) and female elderly (60+) (−0.50, −0.56 to −0.44) showed significantly lower DASS-21 Depression score than male non-elderly (59−) and female non-elderly (59−), respectively (Figure 2).**Chronic disease:** Patients with pulmonary hypertension patients (PHQ-4: 2.30, 1.58 to 3.02), patients with chronic disease (DASS-21 Depression: 8.11, 6.84 to 9.38), and patients with Parkinson’s disease (HADS-Depression: 1.07, 0.13 to 2.01) showed significantly higher depressive symptoms than healthy control. However, there was no significant difference between patients taking immune suppressants compared to healthy control in the HADS-Depression score (−0.22, −1.98 to 1.54) (Figure 2, Appendix A).**Severe mental illness:** Patients with SMI (CES-D: 7.71, 5.70 to 9.72; DASS-21 Depression: 0.37, 0.09 to 0.65) and patients with common mental disorder (DASS-21 Depression: 0.67, 0.42 to 0.92) showed significantly higher depressive symptoms than healthy control. In addition, there was borderline significance between patients with bipolar disorder and healthy control (DASS-21 Depression score: 2.40, −0.08 to 4.89), and patients with psychotic disorder and healthy control (DASS-21 Depression score: −1.19, −2.61 to 0.23) (Figure 2).**Pregnant:** Pregnant women showed significantly higher depressive symptoms than non-pregnant women on DASS-21 Depression (1.47, 0.78 to 2.16), but not on BDI-II (0.79, −0.68 to 2.26) (Appendix A).

#### 3.4.2. Anxiety: Primary Outcome

**Elderly:** Elderly (65+ or 70+) groups generally showed significantly lower anxiety symptoms than non-elderly groups on GAD-7 (−4.37, −5.13 to −3.61 (65+ vs. 18–39 age); −1.80, −2.47 to −1.13 (65+ vs. 40–64 age); −5.20, −5.65 to −4.75 (70+ vs. 18–29 age); −4.00, −4.35 to −3.65 (70+ vs. 30–49 age); −2.00, −2.34 to −1.66 (70+ vs. 50–69 age)) as well as on DASS-21 Anxiety score (−1.74, −2.67 to −0.81 (60+ vs. 18–25 age); −1.69; −2.58 to −0.80 (60+ vs. 26–33 age); −1.25; −2.11 to −0.39 (60+ vs. 34–45 age)). In addition, there was borderline significance between elderly (60+) group and 40–60 age group (−0.79, −1.67 to 0.09) on DASS-21 Anxiety score. Moreover, both male elderly (60+) (−0.40, −0.46 to −0.34) and female elderly (60+) (−0.50, −0.57 to −0.43) showed significantly lower DASS-21 Anxiety score than male non-elderly (59−) and female non-elderly (59−), respectively (Figure 3).**Chronic diseases:** There were no significant differences between patients taking immune suppressants (1.38, −0.31 to 3.07) or patients with Parkinson’s disease (0.27, −0.62 to 1.16) compared to healthy control on HADS-Anxiety score. In addition, children with chronic illness (STAI-S: −2.64, −7.90 to 2.62) or children with cystic fibrosis (STAI-S: −1.00, −5.79 to 3.79) did not show significantly different state anxiety compared to healthy peers. However, children with chronic illness showed significantly higher trait anxiety (STAI-T: 6.23, 0.55 to 11.91) than that of healthy peers. Patients with chronic disease showed significantly higher anxiety symptoms (DASS-21 Anxiety: 7.22, 6.17 to 8.27) than that of healthy control (Figure 3, Appendix A).**Severe mental illness:** Patients with SMI (Patient Reported Outcomes Measurement Information System Anxiety: 3.71, 2.34 to 5.08; DASS-21 Anxiety: 0.85, 0.45 to 1.25), patients with bipolar disorder (DASS-21 Anxiety: 1.87, 0.14 to 3.60), patients with psychotic disorder (DASS-21 Anxiety: 1.38, 0.13 to 2.63), and patients with common mental disorder (DASS-21 Anxiety: 1.46, 1.11 to 1.81) showed significantly higher anxiety symptoms than healthy control (Figure 3).**Pregnant:** There were two conflicting results on state anxiety between pregnant and non-pregnant participants, that one found that that of pregnant women was significantly lower than that of non-pregnant women (STAI-S: −4.66; −7.32 to −2.00), while the other one found no significant difference (STAI-S: 1.15; −1.31 to 3.61). Otherwise, trait anxiety (STAI-T: −3.46, −6.12 to −0.80) and anxiety symptoms (HADS-Anxiety: 0.80, 0.09 to 1.51) of pregnant women were significantly lower than those of non-pregnant women (Appendix A).

#### 3.4.3. Stress—Secondary Outcome

**Elderly:** Elderly (60+) group showed significantly lower stress symptoms than non-elderly groups on DASS-21 Stress score (−3.37, −4.67 to −2.07 (60+ vs. 18–25 age); −2.97, −4.23 to −1.71 (60+ vs. 26–33 age), −2.65, −3.87 to −1.43 (60+ vs. 34–45 age), −1.54, −2.78 to −0.30 (60+ vs. 46–60 age)). In addition, both male elderly (60+) (−1.00, −1.11 to −0.89) and female elderly (60+) (−1.40, −1.52 to −1.28) showed significantly lower DASS-21 Stress score than male non-elderly (59−) and female non-elderly (59−), respectively (Appendix A).**Chronic disease:** Patients with chronic disease showed significantly higher stress symptoms than healthy control on DASS-21 Stress score (8.72, 7.47 to 9.97) (Appendix A).**Severe mental illness:** Patients with SMI (PSS: 1.84, 0.68 to 3.00; DASS-21 Stress: 0.42, 0.94 to 0.10) and patients with bipolar disorder (DASS-21 Stress: 2.91, 0.31 to 5.50) showed significantly higher stress symptoms than healthy control. However, patients with psychotic disorder showed significantly lower stress symptoms than healthy control on DASS-21 Stress (1.98, 3.50 to −0.46). Meanwhile, there was no significant difference in common mental disorder patients compared to healthy control in the DASS-21 Stress (0.19, −0.29 to 0.67) (Appendix A).

#### 3.4.4. PTSD—Secondary Outcome

**Elderly:** Elderly (60+ or 65+) groups generally showed significantly lower PTSD symptoms than non-elderly groups on IES Total score (−0.38, −0.48 to −0.28 (65+ vs. 18–39 age); −0.16, −0.25 to −0.07 (65+ vs. 40–64 age)), IES-R Total score (−5.18, −8.39 to −1.97 (60+ vs. 18–25 age); −5.02, −8.17 to −1.87 (60+ vs. 26–33 age); −4.07, −7.16 to −0.98 (60+ vs. 34–45 age)), IES-R Hyperactivation score (−1.88, −2.86 to −0.90 (60+ vs. 18–39 age); −1.81, −2.77 to −0.85 (60+ vs. 26–33 age); −1.53, −2.47 to −0.59 (60+ vs. 34–45 age)), IES-R Evitation score (−2.10, −3.37 to −0.83 (60+ vs. 18–25 age); −1.48, −2.72 to −0.24 (60+ vs. 26–33 age)), and IES-R Intrusions score (−1.73, −3.15, −0.31 (60+ vs. 26–33 age); 1.37, −2.76 to 0.02 (60+ vs. 34–45 age)). Moreover, both male (60+) and female elderly (60+) showed significantly lower PTSD symptoms than their non-elderly counterparts (IES-Intrusive thoughts: −0.30, −0.41 to −0.19 (male 60+ vs. male 59−); −0.40, −0.52 to −0.28 (female 60+ vs. female 59−); IES-Avoidant Behavior: −0.20, 0.34 to −0.06 (male 60+ vs. male 59−); 0.60, −0.73 to −0.47 (female 60+ vs. female 59−)). A borderline significant difference was found between elderly (60+) and 34–45 age group (−1.17, −2.39 to 0.05) on IES-R Evitation. However, no significant difference was found between elderly (60+) and 46–60 age group (IES-R Total: −1.70, −4.85 to 1.45; IES-R Hyperactivation: 0.90, −1.85 to 0.05; IES-R Intrusions: −0.64, −2.05 to 0.77; IES-R Evitation: −0.15, −1.39 to 1.09). IES-R Intrusions score of elderly (60+) compared to that of 18–25 age group was also not significant (−1.21, −2.66 to 0.24) (Appendix A).**Severe mental illness:** Patients with common mental disorder showed significantly higher intrusive thoughts and avoidant behavior than healthy control on IES-Intrusive thoughts (1.06, 0.62 to 1.50) and IES-Avoidant Behavior score (0.96, 0.52 to 1.40), respectively. In addition, there was borderline significance between SMD patients and healthy control on IES-Intrusive thoughts (0.44, −0.04 to 0.92). However, patients with SMI showed significantly lower avoidant behavior than healthy control (IES-Avoidant Behavior: −0.82, −1.32 to −0.32) (Appendix A).

#### 3.4.5. Sleep—Secondary Outcome

**Chronic disease:** Patients with Parkinson’s disease showed significantly higher sleep disturbance than healthy control (PSQI Global Score: 92.77, 1.85 to 3.69) (Appendix A).

#### 3.4.6. Positive/Negative Affect—Secondary Outcome

**Pregnant:** No significant difference was found between pregnant women compared with non-pregnant women on positive nor negative affect (PANAS: −0.34, −1.67 to 0.99; 0.14, −1.68, 1.98) (Appendix A).

### 3.5. Publication Bias

There was no meta-analysis result including 10 or more studies, therefore, publication bias through funnel plots could not be evaluated.

## 4. Discussion

### 4.1. The Findings

To the best of our knowledge, there was no systematic review and meta-analysis comparing the mental health impact of COVID-19 between the vulnerable and non-vulnerable groups. In this current study, we analyzed psychological impacts of the COVID-19 pandemic among vulnerable populations compared to healthy controls.

Among the 17 studies included in the meta-analysis, 4 studies were on the elderly, 3 on pregnant women, 6 on patients with chronic diseases, and 4 on patients with SMI. In general, the elderly experienced significantly lower severities of several psychological symptoms including depression, anxiety, and perceived stress compared to non-elderly groups. Severity of PTSD symptoms also tended to be significantly lower in the elderly, although several results were non-significant according to the age of comparison groups. Other vulnerable populations including pregnant women, patients with chronic diseases, and SMI showed mixed results according to each mental health outcome. Specifically, the perceived stress level of patients with chronic diseases was significantly higher than that of the control group, whereas depressive symptoms and anxiety were not significant between the two groups. Pregnant women showed mixed results in depressive symptoms and anxiety results, and insignificant results in positive/negative affect. Patients with pre-existing SMI experienced significantly higher levels of anxiety and sleep disturbance compared to the general public, while showing mixed results on depressive symptoms, perceived stress and PTSD symptoms (Table 3). In general, the included studies did not perform random sampling but used convenience sampling, leaving the results susceptible to selection bias. In addition, in most of the included studies, the response rate or drop-out rate of both the total participants and subgroups was not reported, thus it is unclear whether the studies’ validity was appropriately managed.

### 4.2. Clinical Interpretation

Vulnerable populations are known to be affected by pandemic-induced mental impacts to a greater extent than general populations [64]. The pandemic could exacerbate health disparities and increase susceptibility to stress among vulnerable groups [65], which could eventually make them suffer from more depression, anxiety, distress, and post-traumatic stress [39,42,43,47,48,61,66]. Understanding the psychological impact of COVID-19 on different populations would provide clinical implications for the identification of high-risk groups and designing interventions, as well as policies for better mental health care systems, which is of critical importance in the context of public health at a global level.

Previous studies reported that elderly and patients with pre-existing chronic health conditions are not only at a higher risk of infection and mortality from COVID-19, but are also more likely to experience negative psychological consequences [42,48,52,61]. Interestingly, however, the elderly showed fewer symptoms of poor mental health during the COVID-19 epidemic in several included studies. The results of the studies on the elderly could reflect that older people may already have been living alone with few social contacts or may have been relatively free from economic activities before COVID-19, whereas the younger generation may suffer more from the increased psychological impact of economic restrictions or instability—such as lay-offs—caused by the pandemic [51]. Unlike other natural disasters in which social capital is of great importance [16], COVID-19 discourages social contact, so the limited social network of the elderly may have buffered them against the negative impact of COVID-19 on mental health. In addition, relatively high resilience in old age and low levels of psychological well-being in younger ages could be significant factors regarding better mental health outcomes among the elderly [46]. It is consistent with a previous study which found that, compared to younger populations, elderly, more resilient, and risk-averse people experience less anxiety during COVID-19 [67]. Young people adopt acute changes to their learning methods and make use of online-based technologies and devices, which could aggravate psychological distress [68]. Since sudden loss of income and job insecurity due to COVID-19 have been reported to lead to poorer health outcomes [17], the increased financial instability of an economic crisis may result in negative psychological outcomes among young people. These results are consistent with previous research conducted during different pandemics: the influenza outbreak and SARS, during which decreased distress among the elderly was found [69,70]. Further, during mandated “stay-at-home” periods, social distancing increases family time spent at home, which may act as a positive factor for the elderly who had been spending most of their time at home long before the outbreak.

Meanwhile, individuals with chronic health conditions were found to have suffered from more distress compared to healthy controls. Sayeed (2021) found that the perceived stress of chronic disease patients was higher than that of the healthy control. Strict social distancing, quarantine policy, and fear of infections made it difficult for patients with chronic diseases to utilize consistent medical care, particularly in countries where there is a shortage of medical and human resources [42]. Among Parkinson’s disease patients, higher levels of sleep disturbances were found compared to the healthy controls [39]. Difficulties completing ongoing treatments and making the regular visits to health care services that Parkinson’s disease patients need may contribute to the lack of medical consultation and the sleep state of Parkinson’s disease patients. In addition, the incidence of insomnia, anxiety disorders, and depressive disorders may depend on an individual’s pre-existent health conditions—particularly the presence of autoimmune diseases—rather than their profession [71]. Therefore, people with chronic disease during the pandemic require social and clinical support for their psychological well-being and stress reduction.

Greater anxiety responses in SMI patients than general populations indicate that SMI patients carry an intrinsically increased psychological burden [43]. Restricted opportunities due to poor social networks also make it difficult for individuals with SMI to obtain emotional and substantial support from family members or neighbors [72].

Mixed results may have been found owing to different characteristics of participants and study settings. Specifically, the characteristics of the participants could be dissimilar, depending on whether the patient of SMI or chronic disease is an inpatient, an outpatient, or a patient in residential rehabilitation. Patients in rehabilitation communities may perceive more freedom and security compared to those who are in a hospital setting, but may also get more ongoing social support from staff, medical professionals, and peers [55]. Patients with chronic diseases also tend to use more protective actions, including wearing masks, washing hands, and avoiding crowded public areas [41]. In addition, pregnant women have been reported to show different mental health outcomes depending on factors such as age, marital status, and support from their spouse/family members [73].

Mixed results could also be attributed to the different capacity of each country to cope with mental health problems. According to the WHO, there are significant differences between the coping capacities of each country toward mental health problems [74]. Following the discontinuation of face-to-face services, large differences in the acceptance rates of psychiatric interventions adopting telemedicine or teletherapy were reported between high- and low-income countries. The allocation of resources for maintaining core mental health services also varies from country to country. Specifically, 89% of countries from the survey reported that mental health and psychosocial support were part of their country’s COVID-19 coping plans, but only 17% of these countries possessed additional funds to address these issues. Sociocultural factors are also related to mental health problems caused by COVID-19. Cultural background with social values, individualism, avoidant tendency towards uncertainty, and power balance were pointed out to be associated with disparate behavioral responses and mental health [75].

Furthermore, mixed results may depend on different epidemic stages, or when the lockdown policies were enforced in each country, as well as when mental health was measured in each individual study. Previous research with repeated measures reported that while it was not significant for 2–4 days after formal declaration of social isolation, as time passed (after 47 days) depression and anxiety significantly increased among pregnant women [47]. Different mental effects were also found according to the stage of the epidemic: whether it is in its initial stages with a surge of cases and mortality, or in its alleviating stages with an increasing number of cured patients and formal/obvious information/guidelines on COVID-19 [76]. In sum, mental health outcomes were measured at a different time among the included studies, with differences in the timing and duration of lockdown by country.

### 4.3. Strengths and Limitations

There are significant public and clinical implications to the present study. As mental health becomes an increasingly important issue in the era of the COVID-19, identifying the priorities of mental health problems according to each psychological symptom in vulnerable populations may enable efficient use of medical resources. In addition, a comprehensive review of the mental health of the vulnerable in the pandemic will help to develop strategies to preemptively protect the mental health of the vulnerable in a similar pandemic in the future. Taking into account the vulnerability of the subject, the results in this study could be used to develop psycho-social interventions at both clinical and community levels in order to effectively address the threat to mental health posed by COVID-19. Considering the fact that vulnerable groups often lack acceptability and proactive use of medical resources [77], the mental health improvement strategy for the vulnerable group is essential in the area of welfare, and the findings in this systematic review will be helpful in developing welfare strategies for improving national mental health.

There are also certain limitations to the present study. **First**, since we used the first-measured value after the declaration of the pandemic, several results may not have reflected a significant difference throughout time. However, the changing mental health effects over time in the COVID-19 era have been documented in several studies [61,76]. **Second**, our results cannot confirm the prevalence of diagnosed psychiatric disorders, since the included research did not use clinicians’ standardized interviews but only depended on self-reported measures. Given that mental health assessed by self-reported measures may be overestimated compared to mental health assessed by mental health professionals [78], the psychiatric symptoms in both vulnerable and non-vulnerable groups in our findings may have been exaggerated. **Third**, due to the insufficient number and heterogeneity of included studies, we could not obtain the results of a meta-analysis based on a sufficient number of studies on specific psychiatric symptoms in specific vulnerable groups. Therefore, most of the mental health effects were based on observational studies, which could potentially affect the credibility of our findings. **Fourth**, the methodological quality of the included studies was not the best overall. In particular, as most studies did not perform random sampling, the findings of this review may be susceptible to selection bias. This limitation suggests that our findings may be significantly affected by the results published in later high-quality, large-scale observational studies. **Fifth**, the different sociocultural settings of the included studies may have led to the mixed results since this study did not include a sufficient number of studies to analyze cultural factors. The way to respond to disasters and maintain psychological well-being may differ according to one’s cultural background [79], implying that people are affected differently by the COVID-19 pandemic depending on the culture to which they belong. **Sixth**, the geographical location of the participants was not taken into account in our study. However, participants in different geographic areas may have experienced a disparate mental health impact from the pandemic, depending on the severity of COVID-19 exposure in their area. For example, a recent study found that differences of geographical regions and population density may affect the magnitude of negative consequences on mental health outcomes during the COVID-19 pandemic [80]. **Seventh**, we only included comparative studies on ‘a vulnerable population versus a healthy control’ which has left us with large variations in the participants’ age, times from the COVID-19 outbreak, type of vulnerability, and potential comorbid diseases between collected studies. Therefore, in this review, attention was paid to the values of data from individual study rather than pooled data in our meta-analyses. Nonetheless, the variations among included studies could not be eliminated, particularly in terms of age variations in two vulnerable groups—chronic disease patients and SMI patients. **Finally**, despite our detailed search for various vulnerable groups, such as children, those with a disability (disabled), unemployed, low income or in precarious employment, homeless, single parents, racial/ethnic minorities, immigrants, refugee groups, and underinsured (uninsured patient or patients without health insurance), only four vulnerable groups—the elderly, pregnant women, chronic disease patients, SMI patients—were included in the quantitative analysis. Therefore, the mental health impact of COVID-19 on some vulnerable populations could not be assessed. In the future studies the mental health impact of COVID-19 on other vulnerable populations should be examined. Although excluded because they did not meet the inclusion criteria for this review there are also recent studies investigating the mental health of other vulnerable populations, such as racial/ethnic minorities [81] and people with physical disabilities [82]. In the future, in the context of COVID-19, the mental health of these vulnerable populations needs to be investigated more specifically through comparison with an ethnically mainstream or physically healthy control group.

### 4.4. Suggestion of Future Studies

Suggestion for future studies can be made based on the limitations. **First**, future study may wish to examine longitudinal studies on how the mental health of a vulnerable group has changed throughout the COVID-19 era. For example, recently protocols of studies investigating the longitudinal impact of COVID-19 on the mental health of people with disabilities, the elderly [83], and pregnant women and their children [84] have been published. **Second**, the use of clinicians’ standardized assessments other than self-reported measures can be suggested for future studies, by which the prevalence of diagnosed psychiatric disorders can be better confirmed. However, it should be taken into account that face-to-face evaluation is limited in the COVID-19 era. In this context, in Australia, there are ongoing community mental health services, using telehealth modalities such as video conference, online forums, and mobile apps to manage mental health problems [85], which could suggest alternative strategies for clinicians when evaluating mental health in the face of the pandemic. **Third**, as more studies on mental health problems in vulnerable populations become available, future systematic reviews and meta-analyses may be better performed in terms of homogeneity, high methodological quality, and including a sufficient number studies. Regarding mental health issues in the COVID-19 era, there have been many studies investigating the mental health of the general population [86], frontline health care workers [87], and COVID-19 patients [88], but few studies on vulnerable groups. However, given that COVID-19 is disproportionately affecting the physical and mental health of the vulnerable [89], future research which illuminates mental health problems of vulnerable groups should be better supported. **Fourth**, further studies can also be suggested to examine how cultural/geographic factors and the pandemic interact to affect the mental health of each vulnerable group. Since coping strategies and emotional responses to stress situations have been reported to vary among different cultures [90], further studies to confirm the cultural differences in mental health impacts of COVID-19 can be suggested. **Fifth**, future studies may attempt to control the age range of their included studies, or perform subgroup analysis in terms of age, in order to prevent large age variations in the synthesized data of each vulnerable group. In particular, the better mental health of the elderly found in our study could be clarified in further study, through conducting qualitative interviews of the population. **Finally**, we encourage future studies to continue to consider the mental health impact of COVID-19 on other vulnerable populations besides the four vulnerable groups that we examined. To facilitate these studies, consensus may be needed to define ‘mentally vulnerable groups’ in the context of COVID-19.

## 5. Conclusions

Overall, our findings indicate that vulnerable groups have been affected differently in the COVID-19 era. By investigating a wide range of vulnerable groups rather than focusing on a specific vulnerable group, we were able to evaluate whether certain mental health problems were more likely to occur (or be prevalent) in certain vulnerable groups. Although the insufficient number and heterogeneity of included studies leave the results inconclusive, our findings may contribute to identifying the priorities of mental health needs among various vulnerable populations and to allocating health resources efficiently in the present and the future.

## Figures and Tables

**Figure 1 ijerph-18-10830-f001:**
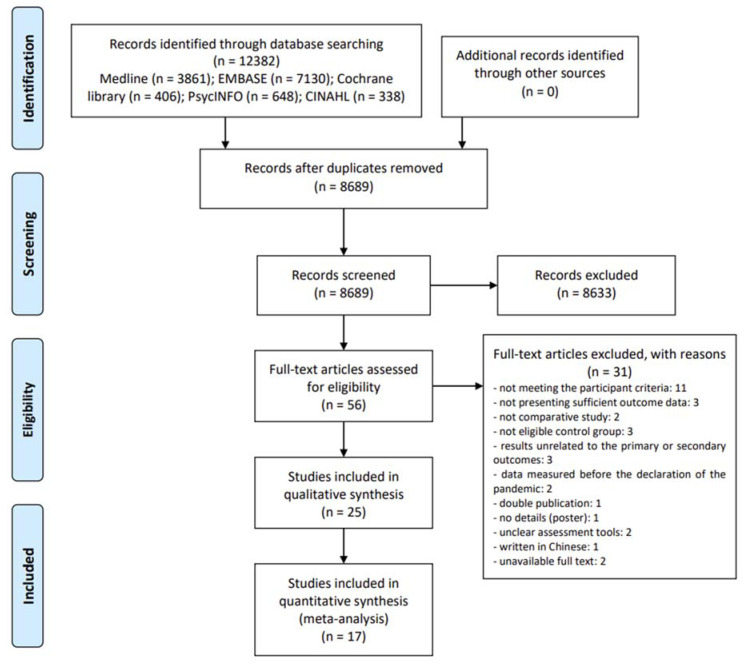
PRISMA Flow diagram.

**Figure 2 ijerph-18-10830-f002:**
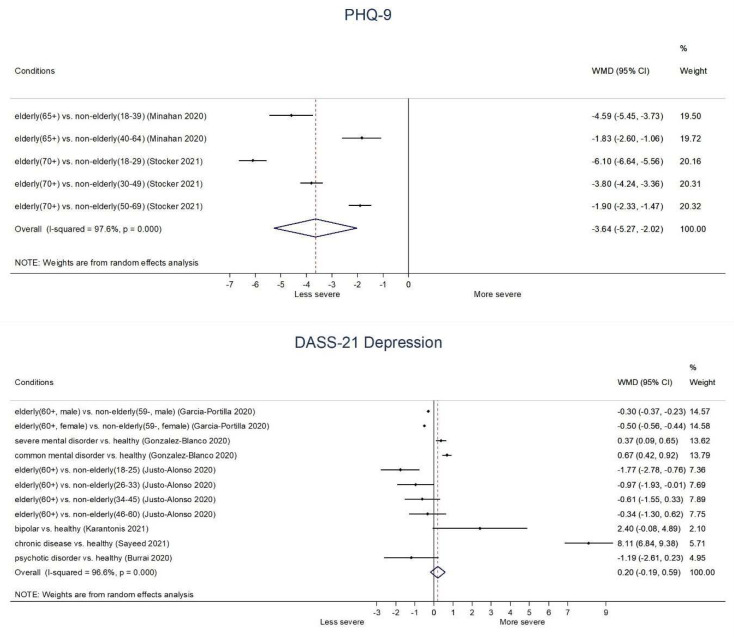
Depressive symptoms in vulnerable groups during COVID-19. **Abbreviations**. DASS-21, Depression, Anxiety, and Stress Scale-21; PHQ-9, Patient Health Questionnaire.

**Figure 3 ijerph-18-10830-f003:**
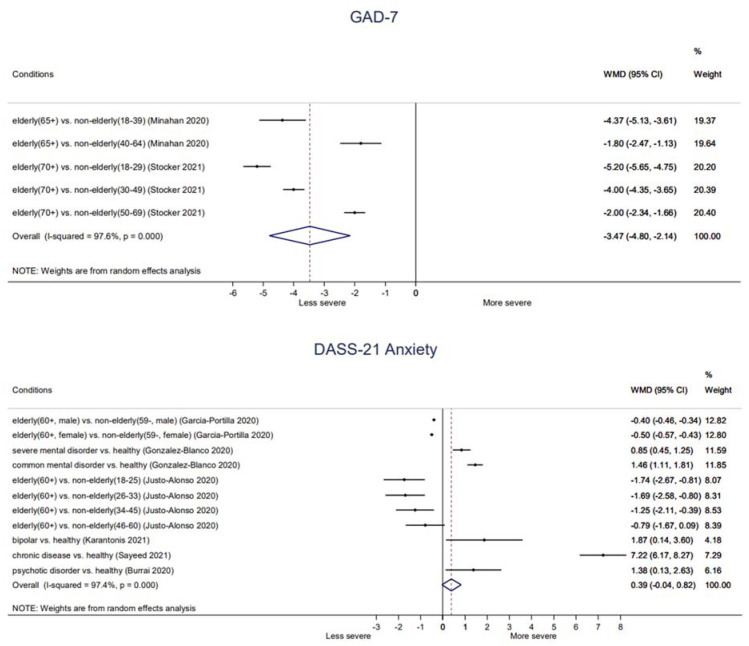
Anxiety symptoms in vulnerable groups during COVID-19. **Abbreviations**. DASS-21, Depression, Anxiety, and Stress Scale-21; GAD-7, Generalized Anxiety Disorder Scale-7.

**Table 1 ijerph-18-10830-t001:** Characteristics of included studies.

Study	Study Design	Country	Ethnicity	Type of Vulnerability	Sample Size (M:F)	Mean Age (Year)	Outcomes	Results
Yassa 2020	prospective CC	Turkey	NR	Pregnant	G1 (pregnant): 203 G2 (non-pregnant):101	G1: 27.4 ± 5.3 G2: 27.6 ± 4.1	1. STAI 20-item	1-1. state anxiety: G1 > G2 ^+^ 1-2. trait anxiety: NS
García-Portilla 2020	CS	Spain	NR	Elderly	G1 (60+): 1690 (831:859) G2 (59−): 13,363 (4308:9055)	G1: 65.9 ± 5.1 G2: (male) 66.5 ± 5.4, (female) 64.4 ± 4.8	1. DASS-21 2. IES	Female/Male 1-1. depression: G1 < G2 ^+^/G1 < G2 ^+^ 1-2. anxiety: G1 < G2 ^+^/G1 < G2 ^+^ 1-3. stress: G1 < G2 ^+^/G1 < G2 ^+^ 2-1. intrusive thoughts: G1 < G2 ^+^/G1 < G2 ^+^ 2-2. avoidant behavior: G1 < G2 ^+^/G1 < G2 ^+^
Cakiroglu 2020	CS	Turkey	NR	Chronic disease	G1 (patients):15 G2 (healthy control): 33	G1: 15.80 ± 2.11 G2: 15.00 ± 2.5	1. STAI 20-item	1-1. state anxiety: NS 1-2. trait anxiety: G1 > G2 ^+^
González-Blanco 2020	CS	Spain	NR	SMI	G1 (SMI): 125 (48:77) G2 (CMD): 250 (96:154) G3 (HC): 250 (96:154)	G1: 43.25 ± 14.41 G2: 43.17 ± 14.27 G3: 43.27 ± 14.37	1. DASS-21 2. IES	1-1. depression: G3 < G1 < G2 ^+^ 1-2. anxiety: G3 < G1 < G2 ^+^ 1-3. stress: G3 < G1 < G2 ^+^ 2-1. intrusive thoughts: G3 < G1 < G2 ^+^ 2-2. avoidant behavior: G1 < G3 < G2 ^+^
Yocum 2021	LS	United States	Multiethnicity	SMI	G1 (BD): 345 G2 (HC): 147	G1, G2: 49	1. PHQ-9 2. GAD-7 3. PSQI	1. depression: G1 > G2 ^§^ 2. anxiety: G1 > G2 ^§^ 3. sleep: G1 > G2 ^§^
Minahan 2020	CS	United States	Multiethnicity	Elderly	G1 (18–39): 375 G2 (40–64): 542 G3 (65–92): 398	G1: 27.98 ± 5.18 G2: 55.44 ± 6.51 G3: 71.32 ± 5.10	1. PHQ-9 2. GAD-7 3. IES	1. depression: G3 < G2 < G1 ^§^ 2. anxiety: G3 < G2 < G1 ^§^ 3. PTSD: G3 < G2 < G1 ^§^
Pinkham 2021	CS	United States	NR	SMI	G1 (SMI): 163 (64:99) G2 (HC): 27 (13:14)	G1: 42.74 ± 11.26 G2: 38.41 ± 12.24	1. CES-D 2. PSS 3. PROMIS-anxiety	1. depression: G1 > G2 ^§^ 2. stress: G1 > G2 ^§^ 3. anxiety: G1 > G2 ^§^
Al-Sofiani 2020	CS	Saudi Arabia	NR	Chronic disease	G1 (patients): 568 (242:326) G2 (HC): 1598 (632:966)	NR	1. PHQ-9 2. GAD-7	1. depression: G1 > G2 ^§^ 2. anxiety: G1 > G2 ^§^
Senkalfa 2020	CS	Turkey	NR	Chronic disease	G1 (patients): 45 (23:22) G2 (HC): 90 (46:44)	Median (IQR) G1: 99.0 mo (63.3–139.5) G2: 106.7 mo (53.0–159.1)	1. STAI-C	1. anxiety: G1 < G2 ^§^
Dadra 2020	retrospective CC	Iran	NR	Chronic disease	G1 (patients): 42 (8:34) G2 (HC): 42 (13:29)	Median (IQR) G1: 32 (24.5–47.75) G2: 37 (32–45.25)	1. HADS	1-1. anxiety: NS 1-2. depression: NS
Justo-Alonso 2020	CS	Spain	NR	Elderly	G1 (18–25): 458 G2 (26–33): 729 G3 (34–45): 1358 G4 (46–60): 749 G5 (60−): 204	G1, G2: 39.24 ± 12.00	1. DASS-21 2. IES-R	1-1. depression: G5 < G4 < G3 < G2 < G1 ^+^ 1-2. anxiety: G5 < G4 < G3 < G2 < G1 ^+^ 1-3. stress: G5 < G4 < G3 < G2 < G1 ^+^ 2. G5 < G4 < G3 < G2 < G1 ^+^
Balci 2021	retrospective CC	Turkey	NR	Chronic disease	G1 (patients): 45 (30:15) G2 (HC): 43 (24:19)	Median (IQR) G1: 67 (60.00–73.50) G2: 66 (58.00–71.00)	1. HADS	1-1. depression: NS 1-2. anxiety: NS
Muro 2020	CS	Italy	NR	Chronic disease	G1 (patients): 1113 G2 (HC): 1125	Range G1: 11–93 G2: 13–85	1. DASS-21	1-1. depression: G1 > G2 ^§^ 1-2. anxiety: G1 > G2 ^§^ 1-3. stress: G1 > G2 ^§^
Xia 2020	CS	China	single ethnicity	Chronic disease	G1 (patients): 119 (61:58) G2 (HC): 169 (76:93)	G1: 61.18 ± 8.77 G2: 59.84 ± 8.15	1. HADS 2. PSQI	1-1. depression: G1 > G2 ^+^ 1-2. anxiety: G1 > G2 * 2. G1 > G2 ^+^
Karantonis 2021	CS	Australia	NR	SMI	Group1 (BD): 43 (19:24) Group2 (HC): 24 (11:13)	G1: 25.3 ± 11.14 G2: 22.79 ± 12.81	1. DASS-21	1-1. depression: G1 > G2 * 1-2. anxiety: G1 > G2 * 1-3. stress: G1 > G2 *
López-Morales 2020	LS	Argentina	NR	Pregnant	Group1 (pregnant):102 Group2 (non-pregnant): 102	G1: 32.59 ± 4.73 G2: 32.54 ± 4.71	1. BDI –II 2. STAI	1. depression: G1 > G2 ^§^ 2. anxiety: G1 > G2 ^§^
Sayeed 2021	prospective CC	Bangladesh	NR	Chronic disease	G1 (patients): 395 (305:90) G2 (HC): 395 (305:90)	G1: 38.37 ± 12.92 G2: 36.17 ± 6.95	1. DASS-21	1-1. depression: G1 > G2 ^+^ 1-2. anxiety: G1 > G2 ^+^ 1-3. stress: G1 > G2 ^+^
Stocker 2021	CS	Australia	NR	Elderly	G1 (18–29): 1337 G2 (30–49): 5148 G3 (50–69): 5897 G 4(70+):1447	NR	1. PHQ-9 2. GAD-7	1. depression: G4 < G3 < G2 < G1 ^§^ 2. anxiety: G4 < G3 < G2 < G1 ^§^
Poll-Franse 2021	retrospective CC	Netherland	NR	Chronic disease	G1 (patients): 4094 G2 (HC): 977	G1: 63.0 ± 11.1 G2: 62.3 ± 13.0	1. HADS	1-1. depression: NS 1-2. anxiety: G1 > G2 ^+^
Dobler 2020	prospective CC	Germany	NR	Chronic disease	G1 (patients): 112 (25:87) G2 (HC): 52 (17:35)	G1: 54.4 ± 14.0 G2: 52.3 ± 8.9	1. PHQ-4	1. depression: G1 < G2 ^+^
Zach 2021	CS	Israel	NR	Elderly	G1 (45–59): 645 (182:463) G2 (60–69): 393 (138:255) G3 (70+): 164 (60:103)	NR	A questionnaire for measuring depressive moods	1. depression: G1 > G2 > G3 ^+^
Burrai 2020	CS	Italy	NR	SMI	G1 (SMI): 77 (51:26) G2 (HC): 100 (50:50)	G1: 46.61 ± 12.81 G2: 46.40 ± 11.52	1. DASS-21	1-1. depression: NS 1-2. anxiety: G1 > G2 * 1-3. stress: G1 < G2 *
Ciprandi 2020	CS	Italy	NR	Chronic disease	G1 (patients): 712 (290:422) G2 (HC): 3560 (1450:2110)	NS	1. CPDI	1. stress: G1 > G2 ^§^
Mirzaei 2021	CS	Iran	NR	Pregnant	G1 (pregnant): 200 G2 (non-pregnant):201	G1: 29.69 ± 5.85 G2: 32.59 ± 6.31	1. HADS	1-1. depression: G1 > G2 ^+^ 1-2. anxiety: G1 > G2 ^+^
Zhou 2020	CS	China	NR	Pregnant	G1 (pregnant): 544 G2 (non-pregnant):315	G1: 31.1 ± 3.9 G2: 35.4 ± 5.7	1.PHQ-9 2. GAD-7 3. PCL-5 4. ISI	1. depression: G1 < G2 ^+^ 2. anxiety: G1 < G2 ^+^ 2. PTSD: G1 < G2 ^+^ 3.sleep: NS

**Note**. *, *p* < 0.05; ^+^, *p* < 0.01; ^§^, *p*-value was not reported. **Abbreviations**. BD, bipolar disorder; CC, case-control study; CES-D, Center for Epidemiologic Studies Depression Scale; CMD, common mental disorders; CS, cross-sectional study; DASS-21, Depression, Anxiety, and Stress Scale-21; G, group; GAD-7, Generalized Anxiety Disorder Scale-7; HADS, Hospital Anxiety and Depression Scale; HC, healthy controls; IES, Impact of Event Scale; IQR, interquartile range; ISI, insomnia severity index; LS, longitudinal study; NS, not significant; PCL-5, The Posttraumatic Stress Disorder Checklist for DSM-5; PHQ-9, Patient Health Questionnaire; PROMIS, Patient Reported Outcomes Measurement In-formation System; PSQI, Pittsburgh Sleep Quality Index; PSS, Perceived Stress Scale; PTSD, posttraumatic stress disorder; SMI, severe mental illness; STAI, State-Trait Anxiety Inventory; STAI-C, State-Trait Anxiety Inventory for Children.

**Table 2 ijerph-18-10830-t002:** The risk of bias of included studies.

Study	Q1	Q2	Q3	Q4	Q5	Q6	Q7	Q8	Q9
Yassa 2020	N	U	U	Y	NA	Y	Y	Y	U
García-Portilla 2020	Y	N	Y	Y	U	Y	Y	Y	U
Cakiroglu 2020	N	U	U	U	U	Y	U	Y	U
González-Blanco 2020	Y	N	Y	Y	U	Y	Y	Y	U
Yocum 2021	N	N	U	Y	U	Y	Y	Y	Y
Minahan 2020	Y	N	Y	Y	U	Y	Y	Y	U
Pinkham 2021	Y	N	U	Y	U	Y	U	Y	U
Al-Sofiani 2020	Y	N	Y	Y	U	Y	Y	Y	U
Senkalfa 2020	N	N	U	Y	U	Y	U	Y	U
Dadra 2020	N	N	N	Y	NA	Y	U	Y	NA
Justo-Alonso 2020	Y	N	Y	Y	U	Y	Y	Y	U
Balci 2021	U	U	N	Y	NA	Y	Y	Y	NA
Muro 2020	U	U	Y	Y	U	Y	U	N	Y
Xia 2020	N	N	U	Y	U	Y	Y	Y	U
Karantonis 2021	U	U	N	Y	U	Y	Y	Y	U
López-Morales 2020	Y	N	U	Y	U	Y	Y	Y	U
Sayeed 2021	Y	N	Y	Y	NA	Y	Y	Y	U
Stocker 2021	Y	N	Y	Y	U	Y	Y	Y	U
Poll-Franse 2021	Y	U	Y	Y	NA	Y	Y	Y	N
Dobler 2020	N	N	U	Y	NA	Y	Y	Y	N
Zach 2021	Y	N	Y	Y	U	Y	Y	Y	U
Burrai 2020	N	U	U	Y	U	Y	Y	Y	U
Ciprandi 2020	Y	N	Y	Y	U	Y	N	Y	U
Mirzaei 2021	U	N	Y	Y	U	Y	Y	Y	U
Zhou 2020	Y	N	Y	Y	U	Y	Y	Y	U

**Note**. Q1, Was the sample frame appropriate to address the target population?; Q2, Were study participants sampled in an appropriate way?; Q3, Was the sample size adequate?; Q4, Were the study subjects and the setting described in detail?; Q5, Was the data analysis conducted with sufficient coverage of the identified sample?; Q6, Were valid methods used for the identification of the condition?; Q7, Was the condition measured in a standard, reliable way for all participants?; Q8, Was there appropriate statistical analysis?; Q9, Was the response rate adequate, and if not, was the low response rate managed appropriately? **Abbreviations**. N, No; NA, not applicable; U, unclear; Y, yes.

**Table 3 ijerph-18-10830-t003:** Summary of results of included studies.

Outcomes (Compared to Control Group)	Elderly	Chronic Disease	Pregnant	SMI
Mental health outcomes				
Depressive symptoms	↓ *	Mixed (↑ *, NS)	Mixed (↑ *, NS)	Mixed (↑ *, ↓ *)
Anxiety	↓ *	Mixed (↑ *, NS)	Mixed (↓ *,↑ *, NS)	↑ *
Stress	↓ *	↑ *	-	Mixed (↓ *,↑ *)
PTSD symptoms	Mixed (↓ *, NS)	-	-	Mixed † (↑ *,↓ *)
Sleep disturbance	-	↑ *	-	-
Positive affect	-	-	NS	-
Negative affect	-	-	NS	-

**Note**. ↑ or ↓, The symptom severity was significantly higher or lower than that of control group; *, *p* < 0.05; †, the results differed between SMI (bipolar disorder and psychotic disorders) and CMD (depression and anxiety) classified by Gonzalez-Blanco (2020). **Abbreviations**. CMD, common mental disorder; NS, not significant; PTSD, posttraumatic stress disorder; SMI, severe mental illness.

## Data Availability

This data used to support the findings of this study are included within the article.

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
