# Peer review of "Comparison of the Mental Health Impact of COVID-19 on Vulnerable and Non-Vulnerable Groups: A Systematic Review and Meta-Analysis of Observational Studies"

_ijerph, 2021, doi:10.3390/ijerph182010830_

Round 1

Reviewer 1 Report

First, I would like to congratulate the authors for their excellent work. Despite being a recent topic, it is very important to identify the most relevant works developed in this area to encourage future studies.

The work is exhaustive, and well documented, the methodology used is appropriate to the type of study in question.

To maximize the strength of the article i suggest the following:

  • In the "Records Excluded" box in Figure 1 there is an extra parenthesis. It should be better explained that of the 25 studies included, 25 are qualitative and 17 are quantitative.
  • The limitations of the study should be better explained.
  • After performing this work, the authors should further explore the topic "suggestion of future studies".

Author Response

  • Response to Comments from Reviewer 1

Overall comment:

First, I would like to congratulate the authors for their excellent work. Despite being a recent topic, it is very important to identify the most relevant works developed in this area to encourage future studies.

The work is exhaustive, and well documented, the methodology used is appropriate to the type of study in question.

To maximize the strength of the article I suggest the following:

Response:              

Thank you for the careful review of our manuscript.

Comment 1:

In the "Records Excluded" box in Figure 1 there is an extra parenthesis. It should be better explained that of the 25 studies included, 25 are qualitative and 17 are quantitative.

Response 1:           

We have deleted the extra parenthesis in the "Records Excluded" box in Figure 1. We included studies that reported quantitative data according to our inclusion criteria. However, since meta-analysis could be performed when homogeneity between studies is secured, not all studies included in our review could be included in meta-analysis. Thus, in the PRISMA flow diagram, all the studies included in our review were written as “Studies included in qualitative synthesis (n=25)” box, and among them, the studies included in meta-analysis are written separately as “Studies included in quantitative synthesis (n=17)”. (Please see Figure 1 in page 5)

Comment 2:

The limitations of the study should be better explained.

Response 2:           

According to your advice, we have additionally elaborated on the limitations of our study in the manuscript. Several suggestions for future studies mentioned in this section (4.3. Strengths and limitations) were deleted and re-added in the next section (4.4 Suggestion of future studies) that we supplemented. (Please see pages 20-21, marked in red)

“The current study also has some limitations. First, since we used the first-measured value after the declaration of the pandemic, several results may have not reflected a significant difference throughout time. However, the changing mental health effects over time in the COVID-19 era have been documented in several studies [61,77]. Second, our results cannot confirm the prevalence of diagnosed psychiatric disorders, since the included research did not use clinicians’ standardized interviews but only depended on self-reported measures. Given that mental health assessed as self-reported measures may be overestimated compared to mental health assessed by mental health professionals [79], the psychiatric symptoms in both vulnerable and non-vulnerable groups in our findings may have been exaggerated. Third, due to the insufficient number and heterogeneity of included studies, we could not obtain the results of a meta-analysis based on a sufficient number of studies on specific psychiatric symptoms in specific vulnerable groups. Therefore, most of the mental health effects were based on observational studies, which could potentially affect the credibility of our findings. Fourth, the methodological quality of the included studies was not overall the best. In particular, as most studies did not perform random sampling, the findings of this review may be susceptible to selection bias. The limitation suggests that our findings may be significantly affected by the results published in later high-quality, large-scale observational studies. Fifth, different sociocultural settings of the included studies may have led to the mixed results since this study did not include a sufficient number of studies to analyze cultural factors. The way to respond to disasters and maintain psychological well-being may differ according to one’s cultural background [80], implying that people are affected differently by the COVID-19 pandemic depending on the culture they belong. Sixth, the geographical location of the participants was not taken into account in our study. However, participants in different geographic areas may have experienced a disparate mental health impact from the pandemic, depending on the severity of COVID-19 exposure in one’s area. For example, a recent study found that differences of geographical regions and population density may affect the magnitude of negative consequences on mental health outcomes during COVID-19 pandemic [81]. Seventh, we only included comparative studies on ‘a vulnerable population versus a healthy control’ which has left us with large variations in the participants' age, times from the COVID-19 outbreak, type of vulnerability, and potential comorbid diseases between collected studies. Therefore, in this review, attention was paid to the values of data from individual study rather than pooled data in our meta-analyses. Nonetheless, the variations among included studies could not be eliminated, particularly in terms of age variations in two vulnerable groups - chronic disease patients and SMI patients. Finally, despite our detailed search for various vulnerable groups, such as children, those with a disability (disabled), unemployed, low income or precarious employment, homeless, single parents, racial/ethnic minorities, immigrants, refugee groups, and underinsured (uninsured patient or patients without health insurance), only four vulnerable groups – the elderly, pregnant women, chronic disease patients, SMI patients - were included in the quantitative analysis. Therefore, the mental health impact of COVID-19 on some vulnerable populations could not be assessed. In the future studies, mental health impact of COVID-19 on other vulnerable populations should be examined. Although excluded because they did not meet the inclusion criteria for this review, however, there are also recent studies investigating the mental health of other vulnerable populations, such as racial/ethnic minorities [82] and people with physical disabilities [83]. In the future, in the context of COVID-19, the mental health of these vulnerable populations needs to be investigated more specifically through comparison with an ethnically mainstream or physically healthy control group.”

Comment 3:

After performing this work, the authors should further explore the topic "suggestion of future studies".

Response 3:           

We have added ‘4.4 Suggestion of future studies’ in 4. Discussion section, according to your advice. (Please see pages 21-22, marked in red)

“4.4 Suggestion of future studies.

Suggestion for future studies can be made based on the limitations. First, future study may wish to examine longitudinal studies on how mental health of a vulnerable group has changed throughout the COVID-19 era. For example, recently protocols of studies investigating the longitudinal impact of COVID-19 on the mental health of people with disabilities, the elderly [84], and pregnant women and their children [85] have been published. Second, use of clinicians’ standardized assessments other than self-reported measures can be suggested for future studies, by which the prevalence of diagnosed psychiatric disorders can be better confirmed. However, it should be taken into account that face-to-face evaluation is limited in the COVID-19 era. In this context, in Australia, there are ongoing community mental health services, using telehealth modalities such as video conference, online forum, and mobile apps to manage mental health problems [86], which could suggest alternative strategies for clinicians when evaluating mental health in the face of the epidemic. Third, as more studies on mental health problems in vulnerable populations become available, future systematic reviews and meta-analyses may be better performed in terms of homogeneity, high methodological quality, and a sufficient number of included studies. Regarding mental health issues in the COVID-19 era, there have been many studies investigating mental health of the general population [87], frontline health care workers [88], COVID-19 patients [89], but few studies on vulnerable groups. However, given that COVID-19 is disproportionately affecting the physical and mental health of the vulnerable [90], future research which illuminate mental health problems of vulnerable groups should be better supported. Fourth, further studies can also be suggested to examine how cultural/geographic factors and the pandemic interact to affect the mental health of each vulnerable group. Since coping strategies and emotional responses to stress situations has been reported to vary among different culture [91], further studies to confirm the cultural difference in mental health impact of COVID-19 can be suggested. Fifth, future studies may attempt to control age range of their included studies or perform subgroup analysis in terms of age, in order to prevent large age variations in the synthesized data of each vulnerable group. Especially, better mental health of the elderly found in our study could be more clarified in further study, through conducting qualitative interview of the population. Finally, we encourage future studies to continue to consider mental health impact of COVID-19 on other vulnerable populations besides the four vulnerable groups that we examined. To facilitate these studies, consensus may be needed to define ‘mentally vulnerable groups’ in the context of COVID-19.”

Reviewer 2 Report

Thank you for submitting your work to IJERPH. I hope the comments below will help in further improving the paper:

1) The definition of a vulnerable population during the COVID-19 pandemic might benefit from further clarification. In particular, you state that individuals younger than 18 years are considered vulnerable. However, the risk of suffering severe illness due to an infection by the SARS-CoV-2 virus is minimal for younger cohorts. In addition, the mortality likelihood is also much lower for younger cohorts than the elderly. So, why do you categorize youths as a vulnerable population? Please clarify.

2) A paper which it seems you have overlooked, but which speaks directly to your paper's topic is the recent work by McCleskey and Gruda (2021; reference below), who use an experimental approach to examine and compare the effects of the COVID-19 pandemic on younger vs older population cohorts. Please include this paper in your meta-analysis or at the very least discuss how this paper relates to your paper's topic and what it might add.

3) Please include higher quality figures (especially Figure 2).

4) Tables should be stand-alone pieces of information, and abbreviations need to be explained in full below each table. This includes the abbreviation SMI across tables. Please ensure to fully define all abbreviations below each table, respectively.

References
McCleskey, J., & Gruda, D. (2021). Risk-taking, resilience, and state anxiety during the COVID-19 pandemic: A coming of (old) age story. Personality and Individual Differences, 170, 110485.

Author Response

  • Response to Comments from Reviewer 2

Overall comment:

Thank you for submitting your work to IJERPH. I hope the comments below will help in further improving the paper:

Response:              

Thank you for your careful review of the manuscript.

Comment 1:

1) The definition of a vulnerable population during the COVID-19 pandemic might benefit from further clarification. In particular, you state that individuals younger than 18 years are considered vulnerable. However, the risk of suffering severe illness due to an infection by the SARS-CoV-2 virus is minimal for younger cohorts. In addition, the mortality likelihood is also much lower for younger cohorts than the elderly. So, why do you categorize youths as a vulnerable population? Please clarify.

Response 1:           

We clarified the vulnerability of young people with references as below. The reason was also added to the revised manuscript.

Although risk of severe illness and mortality caused by the COVID-19 is lower for young people, these populations may be vulnerable in terms of mental health by other factors. For example, school closures at a time when peer relationships are important not only equals loss of education [1], but can also lead to a lack of psychological capitals to preserve their mental health [2]. During the previous Ebola virus epidemic, there were high reporting rates of school dropouts, child abuse/maltreatment, and domestic violence [1]. Economic crisis such as sudden unemployment of their parents may put pressure on the population, which may lead to high levels of stress and mental health problems [3]. Moreover, children who need parental help are more likely to be left alone if their parents are not able to work from home, which made them more vulnerable.

(Please see page 3, marked in red)

“People with chronic diseases, who are particularly suggested as vulnerable individuals in COVID era by Center for Disease Control and Prevention (CDC) [21], patients with serious mental illness, including schizophrenia, bipolar disorder, obsessive-compulsive disorder, major depressive disorder, generalized anxiety disorder, and panic disorder, those with a disability (disabled), the elderly people over 60 years, youth/young people/children under 18 years and pregnant women are included as vulnerable populations. In case of youth/young people/children under 18 years, although the risk of severe illness and mortality caused by the COVID-19 are lower for young people, these populations may be vulnerable in terms of mental health by other factors, including loss of education [17] as well as lack of psychological capitals to preserve their mental health [22], due to school closures. Also, child abuse/maltreatment, domestic violence [17], and unemployment of their parents [23] may affect their mental health status.”

Comment 2:

2) A paper which it seems you have overlooked, but which speaks directly to your paper's topic is the recent work by McCleskey and Gruda (2021; reference below), who use an experimental approach to examine and compare the effects of the COVID-19 pandemic on younger vs older population cohorts. Please include this paper in your meta-analysis or at the very least discuss how this paper relates to your paper's topic and what it might add.

References

McCleskey, J., & Gruda, D. (2021). Risk-taking, resilience, and state anxiety during the COVID-19 pandemic: A coming of (old) age story. Personality and Individual Differences, 170, 110485.

Response 2:           

McCleskey and Gruda (2021) could not be included in our analysis since it did not meet our inclusion criteria that requires a comparative study that compare a vulnerable group with a healthy control. Considering its great relation to our study, however, we have added a potential explanation for our findings regarding the elderly in the discussion section. (Please see page 19, marked in red).

“In addition, relatively high resilience in old age and low levels of psychological well-being in younger age could be the significant factors regarding better mental health outcomes among the elderly [46]. It is consistent with a previous study which found that compared to younger populations, elderly, more resilient, and risk-averse people experience less state anxiety during the COVID-19 [68]. Young people should adopt acute changes of learning methods and use of, online-based technologies and devises, which could aggravate psychological distress [69]. Since sudden loss of income, job insecurity due to COVID-19 have been reported to lead to poorer health outcomes [17], increased financial instability with economic crisis may bring negative psychological outcomes among young people. These results are consistent with previous research during different pandemics; influenza outbreak and the SARS, during which decreased distress among the elderly was found [70,71]. Further, during mandated “stay-at-home” period, social distancing increased family time spent at home, which may act as a positive factor for the elderly who had been spending most of their time at home long before the outbreak.”

Comment 3:

3) Please include higher quality figures (especially Figure 2).

Response 3:           

We attached original figure files with higher quality, including Figure 2.

Comment 4:

4) Tables should be stand-alone pieces of information, and abbreviations need to be explained in full below each table. This includes the abbreviation SMI across tables. Please ensure to fully define all abbreviations below each table, respectively.

Response 4:           

We revised the whole abbreviations of each table, giving full definition (Please see pages 10-11, marked in red).

“Abbreviations. BD, bipolar disorder; CC, case-control study; CES-D, Center for Epidemiologic Studies Depression Scale; CMD, common mental disorders; CS, cross-sectional study; DASS-21, Depression, Anxiety, and Stress Scale-21; G, group; GAD-7, Generalized Anxiety Disorder Scale-7; HADS, Hospital Anxiety and Depression Scale; HC, healthy controls; IES, Impact of Event Scale; IQR, interquartile range; ISI, insomnia severity index; LS, longitudinal study; NS, not significant; PCL-5, The Posttraumatic Stress Disorder Checklist for DSM‐5; PHQ-9, Patient Health Questionnaire; PROMIS, Patient Reported Outcomes Measurement In-formation System; PSQI, Pittsburgh Sleep Quality Index; PSS, Perceived Stress Scale; PTSD, posttraumatic stress disorder; SMI, severe mental illness; STAI, State-Trait Anxiety Inventory; STAI-C, State-Trait Anxiety Inventory for Children.”

Reviewer 3 Report

Dear Authors,

Thank you for the opportunity to read this review. 

This review is well prepared. It meets the requirements for publication in IJERPH and brings new knowledge about mental health in the COVID-19 era. Nevertheless, it is worth adding two important studies relating to a very important group of patients in the COVID-19 era, i.e. patients with autoimmune diseases. Currently, social and clinical awareness of these diseases is poor. Therefore, it is extremely important and valuable for this review to highlight their role in this particular period for the entire medical world

Wańkowicz, P.; Szylińska, A.; Rotter, I. The Impact of the COVID-19 Pandemic on Psychological Health and Insomnia among People with Chronic Diseases. J. Clin. Med. 2021, 10, 1206. https://doi.org/10.3390/jcm10061206

Wańkowicz, P.; Szylińska, A.; Rotter, I. Insomnia, Anxiety, and Depression Symptoms during the COVID-19 Pandemic May Depend on the Pre-Existent Health Status Rather than the Profession. Brain Sci. 2021, 11, 1001. https://doi.org/10.3390/brainsci11081001

All the best

Author Response

  • Response to Comments from Reviewer 3

Overall comment:

Dear Authors,

Thank you for the opportunity to read this review.

This review is well prepared. It meets the requirements for publication in IJERPH and brings new knowledge about mental health in the COVID-19 era. Nevertheless, it is worth adding two important studies relating to a very important group of patients in the COVID-19 era, i.e. patients with autoimmune diseases. Currently, social and clinical awareness of these diseases is poor. Therefore, it is extremely important and valuable for this review to highlight their role in this particular period for the entire medical world

Wańkowicz, P.; Szylińska, A.; Rotter, I. The Impact of the COVID-19 Pandemic on Psychological Health and Insomnia among People with Chronic Diseases. J. Clin. Med. 2021, 10, 1206. https://doi.org/10.3390/jcm10061206

Wańkowicz, P.; Szylińska, A.; Rotter, I. Insomnia, Anxiety, and Depression Symptoms during the COVID-19 Pandemic May Depend on the Pre-Existent Health Status Rather than the Profession. Brain Sci. 2021, 11, 1001. https://doi.org/10.3390/brainsci11081001

Response:              

Thank you for the careful review of our manuscript. The two studies that you mentioned could not be included in our study since they do not meet our inclusion criteria that requires a comparative study that compare a vulnerable group (chronic disease patients) with a healthy control. However, the findings of the two studies relates to our topic and we deeply agree with your comment on the importance of encouraging social and clinical awareness of such chronic disease patients during the COVID-19 era. Thus, we have added interpretations related to the findings of the two studies in our discussion section. (Please see page 19, marked in red).

“Meanwhile, individuals with chronic health condition have found to have suffered from more distress compared to healthy controls. Sayeed (2021) found that the perceived stress of chronic disease patients was higher than that of the healthy control. Strict social distancing, quarantine policy, and fear of infections made patients with chronic diseases difficult to utilize consistent medical care, particularly in countries where they have a shortage of medical and human resources [42]. Among Parkinson’s disease patients, higher levels of sleep disturbances were found compared to the healthy controls [39]. Having difficulties in ongoing treatment and regular visits to health care that Parkinson’s disease patients need may contribute to the lack of medical consultation and the sleep state of Parkinson’s disease patients. In addition, the incidence of insomnia, anxiety disorders, and depressive disorders may depend on an individual’s pre-existent health conditions - particularly the presence of autoimmune diseases - rather than their profession [72]. Therefore, people with chronic disease during the pandemic require social and clinical support for their psychological well-being and stress reduction.”

Reviewer 4 Report

The presented work describes a very complex process of meta-analysis of the results from 25 papers. Methodologically, the process was carried out correctly (the applied methods of statistical data analysis are correct). The main problem, in my opinion, is the unclear selection of groups for comparisons. In my opinion, the method of defining the 4 compared vulnerable groups is flawed.

The study groups, if only in terms of age, are extremely difficult to compare.

These input errors (the statistical analysis process is valid) have negative consequences for the interpretation of the results.

Examples: in the group "chronic diseases" we have a huge variation in the age of patients (I will discuss it only on selected 4 examples of works): Cakiroglu 2020- G1: 15.80 ± 2.11, G2: 15.00 ± 2.5 (we are dealing here with the youth population), and in Xia 2020 G1: 61.18 ± 8., G2: 59.84 ± 8.15, in Muro 2020 - G1: 11- 93, G2: 13-85 (actually the use of DASS21 for examining a child aged 11 is questionable) and additionally in Al-Sofi- nor 2020 (and this is a relatively large group of respondents), there is no data on age ... Similarly, this problem concerns the research of the "Eldery" group: Example: In several studies in this group, comparisons were made between more than 2 groups and here is also the problem of the difference in age groups: Mi-nahan 2020 - G1: 27.98 ± 5.18, G2: 55.44 ± 6.51, G3: 71.32 ± 5.10 and in Justo-Alonso 2020 - G1 (18-25): 458, G2 (26-33): 729, G3 (34 -45): 1358, G4 (46-60): 749, G5 (60-): 204. Unfortunately, there are also gaps in the data on the age of the respondents here! In the SMI group, one group stands out from the rest: Karanto-nis 2021, here we are dealing with young people suffering from SMI. Only the age distribution in the pregnant group does not raise any doubts - all subjects are of a similar age.

In my opinion, the selected works are too diverse in terms of basic variables to be compared and provide reliable knowledge. 

Author Response

  • Response to Comments from Reviewer 4

Overall comment:

The presented work describes a very complex process of meta-analysis of the results from 25 papers. Methodologically, the process was carried out correctly (the applied methods of statistical data analysis are correct). The main problem, in my opinion, is the unclear selection of groups for comparisons. In my opinion, the method of defining the 4 compared vulnerable groups is flawed.

The study groups, if only in terms of age, are extremely difficult to compare.

These input errors (the statistical analysis process is valid) have negative consequences for the interpretation of the results.

Examples: in the group "chronic diseases" we have a huge variation in the age of patients (I will discuss it only on selected 4 examples of works): Cakiroglu 2020- G1: 15.80 ± 2.11, G2: 15.00 ± 2.5 (we are dealing here with the youth population), and in Xia 2020 G1: 61.18 ± 8., G2: 59.84 ± 8.15, in Muro 2020 - G1: 11- 93, G2: 13-85 (actually the use of DASS21 for examining a child aged 11 is questionable) and additionally in Al-Sofi- nor 2020 (and this is a relatively large group of respondents), there is no data on age ... Similarly, this problem concerns the research of the "Eldery" group: Example: In several studies in this group, comparisons were made between more than 2 groups and here is also the problem of the difference in age groups: Mi-nahan 2020 - G1: 27.98 ± 5.18, G2: 55.44 ± 6.51, G3: 71.32 ± 5.10 and in Justo-Alonso 2020 - G1 (18-25): 458, G2 (26-33): 729, G3 (34 -45): 1358, G4 (46-60): 749, G5 (60-): 204. Unfortunately, there are also gaps in the data on the age of the respondents here! In the SMI group, one group stands out from the rest: Karanto-nis 2021, here we are dealing with young people suffering from SMI. Only the age distribution in the pregnant group does not raise any doubts - all subjects are of a similar age.

In my opinion, the selected works are too diverse in terms of basic variables to be compared and provide reliable knowledge. 

Response:              

Thank you for the careful review of our manuscript. We agree with the reviewer's comment.

That is, we recognize the diversity and heterogeneity of the selected works in terms of age variable, particularly in that of the chronic disease patients and SMI patients. Therefore, we described the data analysis method as follows. (“However, the heterogeneity of included studies, including the participants' living environment, times from the COVID-19 outbreak, type of vulnerability, and potential comorbid diseases, were evident. Therefore, in this review, attention was paid to the values of data from individual study rather than pooled data in our meta-analyses.”) And we tried to interpret the meta-analysis values as individually as possible. Nevertheless, we acknowledge that this study cannot be free from the limitations pointed out by the reviewer.

It may be due to our limitation of including only comparative studies comparing ‘a vulnerable group versus a healthy control’ regardless of the participants’ age. Therefore, we have highlighted it as a limitation and described suggestions for future study regarding this issue. (Please see pages 21-22, marked in red).

“4.3. Strengths and limitations

Seventh, we only included comparative studies on ‘a vulnerable population versus a healthy control’ which has left us with large variations in the participants' age, times from the COVID-19 outbreak, type of vulnerability, and potential comorbid diseases between collected studies. Therefore, in this review, attention was paid to the values of data from individual study rather than pooled data in our meta-analyses. Nonetheless, the variations among included studies could not be eliminated, particularly in terms of age variations in two vulnerable groups - chronic disease patients and SMI patients…”

“4.4 Suggestion of future studies.

Fifth, future studies may attempt to control age range of their included studies or perform subgroup analysis in terms of age, in order to prevent large age variations in the synthesized data of each vulnerable group. Especially, better mental health of the elderly found in our study could be more clarified in further study, through conducting qualitative interview of the population. Finally, we encourage future studies to continue to consider mental health impact of COVID-19 on other vulnerable populations besides the four vulnerable groups that we examined. To facilitate these studies, consensus may be needed to define ‘mentally vulnerable groups’ in the context of COVID-19.”

Round 2

Reviewer 4 Report

Well done correction. The paper is intersting and deserves to be published in its current form.